# Next Generation Sequencing Technology in the Clinic and Its Challenges

**DOI:** 10.3390/cancers13081751

**Published:** 2021-04-07

**Authors:** Lau K. Vestergaard, Douglas N. P. Oliveira, Claus K. Høgdall, Estrid V. Høgdall

**Affiliations:** 1Molecular Unit, Department of Pathology, Herlev Hospital, University of Copenhagen, DK-2730 Herlev, Denmark; lau.kraesing.vestergaard@regionh.dk (L.K.V.); douglas.nogueira.perez.de.oliveira@regionh.dk (D.N.P.O.); 2Juliane Marie Centre, Department of Gynecology, Rigshospitalet, University of Copenhagen, DK-2100 Copenhagen, Denmark; claus.hogdall@regionh.dk

**Keywords:** bioinformatic pipeline, cancer, next-generation sequencing, alignment, variant calling, clinical application

## Abstract

**Simple Summary:**

Precise identification and annotation of mutations are of utmost importance in clinical oncology. Insights of the DNA sequence can provide meaningful knowledge to unravel the underlying genetics of disease. Hence, tailoring of personalized medicine often relies on specific genomic alteration for treatment efficacy. The aim of this review is to highlight that sequencing harbors much more than just four nucleotides. Moreover, the gradual transition from first to second generation sequencing technologies has led to awareness for choosing the most appropriate bioinformatic analytic tools based on the aim, quality and demand for a specific purpose. Thus, the same raw data can lead to various results reflecting the intrinsic features of different datamining pipelines.

**Abstract:**

Data analysis has become a crucial aspect in clinical oncology to interpret output from next-generation sequencing-based testing. NGS being able to resolve billions of sequencing reactions in a few days has consequently increased the demand for tools to handle and analyze such large data sets. Many tools have been developed since the advent of NGS, featuring their own peculiarities. Increased awareness when interpreting alterations in the genome is therefore of utmost importance, as the same data using different tools can provide diverse outcomes. Hence, it is crucial to evaluate and validate bioinformatic pipelines in clinical settings. Moreover, personalized medicine implies treatment targeting efficacy of biological drugs for specific genomic alterations. Here, we focused on different sequencing technologies, features underlying the genome complexity, and bioinformatic tools that can impact the final annotation. Additionally, we discuss the clinical demand and design for implementing NGS.

## 1. Introduction

Insights into the sequence of DNA can provide meaningful knowledge to unravel the genetics of disease. This approach has propelled diagnostic and treatment strategies to a new level, where personalized medicine is gradually becoming adopted in the clinic. The advent of second-generation sequencing technologies, also known as next-generation sequencing (NGS), has contributed remarkedly with the demand for more economical and faster sequencing technologies. NGS performs massive parallel sequencing and is steadily replacing its predecessor, the traditional Sanger sequencing (Sanger et al., 1977) [1]. Its technologies have made it possible to resolve billions of sequencing reactions in few days from library preparations to end results. Nonetheless, the handling of such substantial amount of data poses a current challenge regarding their interpretation in a clinically meaningful way. Hence, that demand was shortly followed by the development of a plethora of devised NGS bioinformatic tools, each serving its own purpose. Most computational tools such as Bowtie2 [2], Burrows–Wheeler Aligner (BWA) [3], Mutect2 [4] and Strelka2 [5] are freely available for processing NGS data in the scopes of (i) sequence mapping, (ii) base calling and (iii) variant calling. The application of different tools have shown to vary in consistency [6,7], highlighting the necessity of caution and experience, as the output could lead to misguidance of diagnosis, prognosis and personalized treatment in a clinical setting. In this review, we focused on the many factors that influence data interpretation and its application in oncology. This covers sequencing technologies, data output from sequencing, pitfalls and bioinformatics concerns. Finally, we discussed the increasing clinical demand for the implementation of NGS. 

## 2. Sequencing Technologies

Sequencing is the process of determining the order of Adenine (A), Guanine (G), Cytosine (C) and Thymine (T) bases, which makes up the primary structure of DNA. The first two DNA sequencing methodologies are known as Maxam–Gilbert sequencing—a chemical approach [8]—and Sanger sequencing—a chain termination approach [9]. The clinical utility of Maxam–Gilbert sequencing is unknown; hence the latter will be further addressed herein. Sanger sequencing provided the basis for The Human Genome Project [10] given its accuracy, robustness and simplicity [11]. Briefly, the method is based on 4 polymerase chain reactions (PCR), where in each reaction on of the nucleotides is incorporated by a specific fluorescent chain-terminating dideoxynucleotide (ddNTP). The ddNTP incorporation during the in vitro DNA replication is random, producing fragments with varying length. Subsequently size separation via gel electrophoresis reveal the arrangement of the nucleotides based on where the fragment was terminated [9]. The specific fluorescence embedded in each ddNTP (ddATP, ddGTP, ddCTP, ddTTP) allows to read and annotate the sequence. Despite being a very robust and precise method, the Sanger sequencing can only be performed for a single target at a time. Hence, assessing even a small panel of targets makes this approach cost- and time-inefficient. 

The application of faster and lower cost sequencing was introduced by the NGS technologies. Such platforms, as the Illumina and Ion Torrent, are predominantly being used in clinical settings. These two platforms are different in their underlying chemistry of determining the base sequences. In Illumina platforms, a library is first prepared (DNA templates with barcodes and adaptors attached), followed by denaturation to single strands and immobilization on a flow cell. Next, the templates are amplified to form clusters of clonal fragments by bridge amplification [1]. That step is important to yield enough signal for detection during sequencing. The sequencing methodology of Illumina is based upon cyclic reversible dye chain termination in which ddNTP contains different cleavable fluorescence and a reversible blocking group. For each round of sequencing (1) a nucleotide is added to the flow cell; (2) the fluorescent signals are captured and converted into A, T, C or G; (3) the blocking groups are removed; and (4) the process is repeated with a new round of nucleotide incorporation until the strands are synthesized. The Ion Torrent sequencing technology is based on the incorporation of a DNA template from a prepared library together with a single bead into a droplet, referred as bead emulsion. Each reaction unit allows for emulsion PCR to clonally amplify the template until it covers the entire surface of the corresponding bead. This step is analogous to Illumina’s flow cell, it is yielding sufficient signal during sequencing. The covered beads are loaded onto a chip constructed with millions of micro wells to harbor a single bead in each. The sequencing process is based on pH. Each time a nucleotide is added to the synthesizing template a H^+^ (hydrogen ion) is released and detected as a change in pH, the chip is flooded with one nucleotide at a time and the process is repeated hundreds of times. 

For the past few years, a new generation of sequencing methods has been under development. In contrast to NGS technologies generating short reads, third generation sequencing (TGS) aims to generate long reads up to 30.000 bases (30 kb) in length in real time [12]. The MinION from Oxford Nanopore Technologies (ONT) and the single molecule real-time (SMRT) technology from Pacific Bioscience (PacBio) are two types of a TGS technology. TGS bypass the prerequisite for DNA amplification underlying NGS technologies [11]. Hence, avoiding the inherited errors from the amplification step and creating a fast transition from sample acquisition to sequencing. Yet, the sequencing error rate is still high, 10–15% for SMRT [13] and 5–20% for MinION [13] challenging its utility in a clinical setting. A brief comparison of the technologies is presented in Table 1. 

## 3. Extend of Sequencing

The human genome is constituted by approximately 3 billion nucleotides, of which around 1% encode for protein-coding genes [14]. Mutations introduced into these genes may show consequences by misfunctioning (loss-of-function) or dysregulation (gain-of-function) of proteins crucial for homeostasis, leading to cancer. Mutations are defined as driver mutations when acquiring a cellular phenotype that contributes with an advantage of proliferation and/or survival [15]. Besides driver mutations, the genome is lodging thousands of mutations that are randomly dispersed throughout the genome. These are referred to as passenger mutations and exhibits no immediate phenotype and/or beneficial advantage [16]. In 80% of the cases, cancer is a multifactorial and non-mendelian disease, with somatic mutations found in associated genes at disease [17]. In the remaining 20%, germline mutations are identified [17]. Identification of rare events in genes contributing to tumorigenesis is important for the ongoing understanding of cancer [18]. NGS has led to the discovery of numerous candidates associated with cancer [19]. Noteworthy, detection of variants via bioinformatic methods can only prioritize novel findings of mutations and genes for functional testing. Hence, can only mutations as drivers of tumorigenesis, which needs validation on experimental settings [20]. Thus, bioinformatic tools should be perceived more as a predictor than a validator.

NGS enables the generation of data from a full genome (Whole Genome Sequencing, WGS) in a few working days. Sequencing of the protein-coding genes (Whole Exome Sequencing, WES) and exome sequencing of selected genes alone or in combination with hot spots regions (Targeted Exome Sequencing, TES and Panel sequencing, PS) has also become available. In clinical oncology, the latter approach is employed extensively for its ability to target cancer related gene-panels with a fast response time. The broad scope of WGS and WES exhibits some challenges to be implemented into clinical setting (discussed later). Thus, to provide information of diagnostic classification, guide therapeutic decisions and/or enlighten prognostic of tumor in shorter time, assessment of gene panels is an informative approach.

## 4. Targeted Drug Therapies

Several cancer therapies rely on a certain genomic profile to obtain treatment efficacy. Therefore, precise detection of mutations is critical. Sanger sequencing has until recently been used in diagnostic, despite being restricted to few genes. Hence, providing oncologists with limited information about the tumor mutational profile, leading to a more one-fits-all type of therapies [21]. However, given its massive generation of information, NGS promoted the foundation for targeting disease based on individual genomic profile, referred to as Personalized Medicine or Precision Medicine (PM). This concept gives the opportunity for an accurate and effectively treatment strategy [22].

Precise annotation of mutations is required to transform staggering amount of sequencing data into clinically relevant variants with high confidence. Hence, pushing optimal tailoring of a therapeutic course. For instance, poly (ADP-ribose) polymerase (PARP) inhibiting drugs, is used in managing patients with ovarian cancer [23] and breast cancer [24] in cases of pathogenic *BRCA-1/2 -*gene mutations.

Additionally, Imatinib, a small molecule that competitively binds to the active site of a tyrosine kinase, is used in treatment of gastrointestinal stromal tumors in cases of c-*KIT* gene mutations [25]. Moreover, competitive kinase inhibitors, are directed towards the increased activity of *BRAF* induced by a specific alteration (V600E/K) in the *BRAF* -gene in patients diagnosed with malignant melanoma [26]. *KRAS* mutations are observed in 15–25% of all cancers, whither 30–40% of colorectal cancer harbor at least a single mutation on that gene [27]. On the other hand, epidermal growth factor receptor (EGFR)-inhibitor, targeting the EGFR on the surface of cell is used in cases of colorectal cancer without mutations in the *RAS-*genes [28]. A selection of specific genetic alterations to guide treatment is presented in Table 2.

### Additional Annotation Tool—Drug Databases

Many variants have well-established clinical relevance with targets for molecular therapy. Nevertheless, guiding treatment decisions for novel or rare somatic mutations might be challenging. In that regard, growing databases of variants and putatively beneficial drug molecular targeting are in constant development and can be useful tools to assist guidance, such as the Catalogue Of Somatic Mutations In Cancer (COSMIC) [29,30], ClinVar [31,32] and Precision Oncology Knowledge Base (OncoKB) [33]. Contents of these databases are derived from in vitro and/or in vivo validation studies and clinical investigation expert panels. With the incorporation of NGS into the clinic, such databases are steadily growing and improving as means to assist the treatment of future patients.

## 5. Precautions of Data Output from Sequencing

The overall demand for sequencing is to annotate accurate mutations, such as single nucleotide variants (SNV), insertions/deletions (indels), copy number variation (CNV) and structural variation (SV). That should be acquired ideally with high sensitivity (true positives) and specificity (true negatives). A general principle of sequencing is that the broader the scope the lesser the read depth. WGS is on average sequenced to depths of 30–50x [34], making it more explorative oriented, but efficiently enough to identify most germline mutations including, SNV and indels. It also allows for a comprehensive large scale genomic detection of relevant variants, such as large SV or CNV across the whole genome [35]. However, WGS may be insufficient in detecting rare somatic mutations harboring a cancer genome.

The therapeutics available today are exclusively directed against pathogenic alterations in the coding genome. Thus, knowledge of mutations in intronic regions are less informative in clinical oncology. In that regard, WES or large gene panels are more suited for this purpose, where regions can reach an average coverage of 200x [34]. However, targeted sequencing focus on sequencing regions of choice, often gene panels associated with cancer in specific organs with clinical impact. By narrowing the scope to a selected panel, genes and hot-spot regions can be sequenced to depths with more than 1000x coverage [34]. That entails its capability to reach depth able to detect unique low-frequency allele somatic mutations.

NGS is a multifactorial technology, and wariness is important when interpreting results. Factors that may influence results include; type of biological specimen; preanalytical treatment; pseudogenes and repetitive regions; bioinformatic challenges dealing with alignment and variant calling.

### 5.1. Type of Biological Specimen

Biological specimens vary according to whether the material is collected in a research facility or in a clinical setting. Research facilities most often deal with cultured cells and/or xenograft models leaving high quality DNA to be subjected for sequencing. When human tissue is collected from biopsy or radical surgery in a clinical setting, it is commonly subjected to formalin fixation and embedded into paraffin blocks (FFPE) or may optimally be collected as fresh frozen (FF) tissue.

#### 5.1.1. FF and FFPE Tissue

At pathology departments tissues are routinely FFPE-prepared, which are stored at room temperature and confers more flexibility in applications. FFPE samples are used in histopathological examination, immunohistochemistry and/or in situ hybridization. Thus, allowing for further molecular characterization of the tumor. However, the chemical procedure underlying FFPE samples facilitates significant fragmentation and chemically artificial modifications to the nucleic acids, altering the DNA sequence [36]. Formalin-fixation is the primary cause of fragmentation, degradation and incorporation of alterations caused by deamination resulting in C:G > T:A transitions [37]. Hence, these modifications can mislead interpretation of NGS results and potentially guide an inaccurate therapeutic course in a clinical setting. A recent study from Gao and colleagues reports a high mutational concordance comparing FF and FFPE in an NGS multi-gene panel [38]. However, some mutations are introduced due to the higher level of false-positive variants. Kerick and collaborators further reported that one strategy to deal with FFPE artefacts is to increase the sequencing depth [39]. Hence, the increment of depth decreases the number of false positives and false negatives. More stringent alignment filtering is another option, accommodating removal of putative false positives calls with lower quality scores, but this approach will on the other hand compromise read depth.

#### 5.1.2. Liquid Biopsies

Liquid biopsy is another preparation used for clinical NGS. Whole blood collection is a non-invasive procedure and has been used for supporting diagnostics and/or monitoring circulating biomarkers. For instance; cancer antigen 125 (CA-125) for ovarian cancer [40]; CA-11-19 for colorectal cancer [41], CA-19-9 for lung cancer [42], and CA-15-3 for breast cancer [43], are well established markers in clinical practice. Circulating tumor DNA (ctDNA), tumor-derived cell-free DNA, may be promising in diagnosis of cancer and/or monitoring of relapse or progression [44]. Hence, also applicable for guiding therapeutic and monitoring in patients with known cancer. Elevated levels of ctDNA is present in plasma of cancer patients [45]. Nonetheless, the amount is still only a fraction in the pool of circulating cell-free DNA, challenging the current utility of ctDNA as a biomarker. Noteworthy, detection of somatic mutations in ctDNA for application of diagnosis of cancer, supportive guidance for optimal treatment strategy, and for surveillance of progression or recurrence is extensively under investigation [45]. ctDNA enters the plasma due to apoptosis and/or necrosis. A hallmark of apoptosis is the cleavage of DNA orchestrated by activated caspase activity. A study from Mouliere and collaborators, observed that the fragment size across 18 cancer types showed enrichment of ctDNA in lengths shorter than 167 bp and a notably enrichment ranging from fragment from 250 bp to 320 bp in size [46]. Analysis of ctDNA requires highly sensitive techniques for its detection an enrichment owing to the relative low fraction of tumor DNA dispersed within background levels of normal circulating free DNA [47]. The examination of ctDNA from liquid biopsies may be an alternative in the management of metastatic cancers, where no tumor tissue can be obtained.

### 5.2. Homopolymers, Repetitive Regions and Pseudogenes

The genome harbors areas that are difficult to interpret, due to the presence of homopolymers, G/C rich regions, repetitive regions and pseudogenes [48]. This results in substantial differences concerning sequencing depth and the uniformity in sequencing coverage, making these regions difficult for alignment and variant calling [49]. For instance, homopolymer regions are a challenge for the Ion Torrent sequencing platform and it introduces systematic errors due to loss of resolution above 6 nucleotides, for which may cause mis-alignments [21]. Regions with increased G/C content can be lost and subsequently often observed as higher background due to their ability to form secondary structures [48], thus affecting the uniformity of sequencing coverage in these regions. Repetitive regions are widespread in the genome and encode tandemly repeated or close-identical sequences of variable length, often located in regions of introns [50]. Thus, being a hotspot-entry for genomic rearrangement. The concern regarding repetitive regions is to deal with alignment uncertainties due to reads that subsequently align to multiple regions, instead to an unique location. The multi-alignment reads are an obstacle that may affect variant calling as it can originate from multiple sites. Another type of structural feature embedded in the genome is pseudogenes derived from gene duplications. Pseudogenes are sequences that resembles their protein-coding counterparts with high similarity. Those are however, non-functional due to impairing mutations [51]. Albeit non-functional, some pseudogenes are transcriptionally active and act to regulate their parent protein-coding gene through the microRNA pathway [52,53]. Reads from a desired region of interest might have decreased mapping-quality, due to the presence of a pseudogene homolog causing reads to be mis-aligned to the pseudogenes or vice versa. Some clinically relevant genes may encounter pseudogenes, such as *KRAS* [53] for colorectal cancer and *BRCA1* [54] for ovarian cancer and breast cancer. A targeted strategy is therefore required to avoid interference from their pseudogenes that might challenge the interpretation during NGS analyses.

### 5.3. Bioinformatics

A large number of bioinformatic tools have become available in recent years with the aim to navigate and handle the large quantity of raw data generated by the NGS technology [55].

Raw reads from NGS platforms undergo several bioinformatic processes including base calling; quality check; adaptor trimming; read alignment and post processing; variant calling; and finally, variant annotation for functional interpretation of results. An overall of the bioinformatic pipeline available for analyzing NGS is shown in Figure 1. The two most prominent bioinformatic processes that might influence the final interpretation are the tools used for alignment and variant calling [6,7]. Both are numerous and diverse in their underlying algorithms as their original design can often be reserved for a specific purpose, such as WGS, WES, or TES/hot-spot [56]. The challenge associated with artefacts from the material used, library preparation, sequencing technologies and regions selected for sequencing, all underscore the importance of selecting appropriate benchmark tools for specific aims. The different structural peculiarities of the four groups of genomic alterations (SNV, indels, CNV, SV), excludes the possibility of one versatile tool for identifying all variants within the four groups [48]. Improper alignment to the reference genome can significantly constitute discovery of false positive and/or exclusion of disease relevant variants in downstream analyses. During the years, identification of discordance between aligned reads and the reference genome has greatly improved due to progression of variant callers and their ability to handle large amounts of data [57]. However, calling SNVs still remains a challenge after all, as various tools can result in a divergent outcome [6].

#### 5.3.1. Alignment

Bioinformatic approaches are centered around alignment. Variants refer to identifying deviations from a non-cancerous normal reference genome. Alignment to a reference genome is a prerequisite for optimal analysis of NGS data. In that regard, for the human genome there are essentially two main reference builds currently employed: (a) The Genome Reference Consortium Human build 37 (GRCh37 or hg19), published in 2009; and (b) GRCh38 (hg38), released in 2013. The latter is built on data from many donors, subsequently altering 8000 SNV, correction of misassembled hard accessible region, filled-in gaps and added sequences for centromeres [58]. The GRCh38 improvements over GRCh37 have been reported by Pan et al., to give a more accurate genomic analysis results [59]. Additional studies from Guo et al. and Kumaran et al., examined 30 WES data sets examined WES data from NA12878 [6,58], respectively. In both studies they concluded better accuracy and performance using hg38 as reference genome. It has been reported that aligners can affect the variant calling, when dealing with low quality base scores [57,60].

A list of alignment tools is presented in Table 3. The alignment tools Bowtie2 [2] and BWA [3] are known as the two most efficient aligners to date, with most studies using BWA-MEM (maximal exact matches) as the preferred alignment tool [6,57,59,61,62,63,64,65,66,67]. A study from Yu and collaborators investigated the three full-text index in minute space (FM)-indexing aligners (Bowtie2, BWA-MEM, SOAPv2) and one hash-table algorithm aligner (Novoalign) [60]. Preconditions with relatively good base-quality showed similar performance on alignment [60]. However, removal of low quality base improved the alignment performance for Novoalign [60]. The quality of called bases might significantly impact the performance of aligners, where assessment of Phred scores can support choosing the best fitting alignment tool (FM-index- or Hash-table-based algorithm).

Briefly explained, FM-index alignment tools are derived from the Burrows-Wheeler Transform [68]—a method to sufficiently compress large amount of data and finding approximate matches of sequences in the reference genome [69]. Hash table-based aligners uses the seed-and-extend method in combination with additional alignment algorithms [68,70,71].

**Table 3 cancers-13-01751-t003:** Short read alignment tools.

Alignment Tools	Model	Latest Version	Ref.
Bowtie2	FM-index	v2.4.2	[2]
BWA-MEM	FM-index	v0.7.17	[3]
CUSHAW3	FM-index	v3.0.3	[72]
GSNAP	Hash-table	N/A	[73]
ISAAC	FM-index	v4	[74]
MOSAIK	Hash-table	v2.6.0	[75]
Novoalign	Hash-table	v4.03.01	http://www.novocraft.com/ *
SOAPv2	FM-index	v2.20	[76]

* accessed date: 6 April 2021.

Thus far, short-read alignment tools are commonly challenged by encountering reads that maps to multiple locations in the reference genome [50]. Hence, 3 strategies are proposed to deal with multi-reads [50]. First being to discard all multi-reads leaving only unique mapped reads to be processed. However, this strategy would consequently leave out reads with repetitive regions and gene-families, putatively harboring significance. Second is the best matching strategy, reporting reads to the location(s) with the smallest number of mismatches. The third being to report all reads and their location(s) up to a desired threshold.

#### 5.3.2. Variant Calling for SNV and Small Indels

Many tools covering SNV detection have likewise been developed. A list of supported variant-calling tools is presented in Table 4. Consequently, their underlying algorithm of error models and assumptions for identifying mutations result in diverse variant calling across tools [67]. Hence, the methods used for variant calling are an important factor that influence mutational calling when aiming at high sensitivity and specificity. Studies subjecting sample NA12878 [6,57,59,61,62,63,64,65,66,67], shows that GATK-Haplotype Caller (GATK-HC), Mutect2, SAMtools and Strelka2 are among the best performing variant callers for identifying SNV and small indels [6,57,59,61,62,63,64,65,66,67]. The study of Chen et al., additionally revealed that Strelka2 showed better variant calling and sensitivity with a mutation frequency of ≥20% whereas Mutect2 performed better at ≤10% [62]. Strelka2 was developed to be fast and accurate in calling somatic variations [5]. A fast resolution time encounter an important aspect when employed in clinical oncology. Strelka2 showed to be 18–22 times faster than Mutect2 when processing 100–800x WES samples [62]. Nevertheless, a comprehensive comparison evaluating GATK-HC, Mutect2, SAMtools and Strelka2 against each other remains to be elucidated. An overview of tools and their combination in these studies are shown in Table 5.

#### 5.3.3. Variant Calling for CNV and SV

Both CNVs and SVs are commonly found associated with cancer incidences [100], [101]. CNVs covers somatic structural changes of amplification and/or deletion of DNA regions in a chromosomal region [48]. *ERBB2 (HER2)* is an example of a gene often associated with increased copy-number in breast cancer and clinical relevance for detection [102], whilst *TP53* variants are often observed as loss of the wildtype allele [103]. SVs covers structural changes in terms of large translocations and chromosomal rearrangements [48]. Creation of known and novel tumorigenic fusion proteins as well as de-positioning the proximity of regulatory elements for mRNA transcription might indirectly affect cell function contributing to cancer [104].

Tools for identifying SNVs and small indels are not suited for calling variants of CNVs and SVs. A number of tools exists for calling CNV and SV, as shown in Table 4. However, certain challenges are peculiar to those chromosomal changes. Moreover, the technological limitations of short reads generated by NGS (~150 bp) is not sufficient to resolve long insertions and duplicated regions [105]. Thus, the ongoing development of TGS technologies will contribute to unravel CNVs and SVs more accurately. Implementation of the variant allele frequency may benefit to provide hints of CNV and SVs as the variant allele frequency will increase or decrease with the number of copies [106].

The variant calling for CNV and SV harbors different strategies to identify modifications including read-pairing, read-depth, split-read and read-assembly. Read-pairing is the detection of which read pairs are aligned with increased or decreased distance and/or orientations [105]. This method is largely dependent on the insertion size, as small insertion can be ignored or missed by the algorithm [107].

The read-depth method assumes a Poisson distribution in the depth of aligned reads and examines the distribution of reads to reveal duplications and/or deletions. Thus, duplicated regions show increased read depth whereas deleted regions show decreased read depth compared to normal diploid regions [108]. Mapped reads with low confidence in regions of repetitive DNA challenges the accuracy of copy-number status and may introduce a biased output [107].

The split-read approach utilizes read pairs to define breakpoints of structural variants. The concept of the method relies on reads that align with high confidence to the reference. Hence, the unaligned read(s) may potentially define the breaking point of the insertion [109]. However, the split-reads approach shows limitation, as reads below 1000 bases affect both sensitivity and specificity [109]. Finally, the read-assembly method is in theory the most versatile approach for identifying variants. As the method suggests, it is based on assembling a read base scaffold genome that is subsequently compared to the reference genome to identify variants [109]. Nevertheless, the method requires a significant demand on computational resources and longer-length reads, hence it is not advantageous used for the detection of CNV and SV, yet.

## 6. Clinical Demand

NGS assistance to guide diagnostics, prognostics and improve precision medicine are being progressively adopted into the clinic. Furthermore, genomic research has become an area of impact to prioritize mutations for functional testing. Thus, contributing to reveal mechanisms and better understanding of cancer, allowing for the development of new targeted drugs. In clinical oncology, attention to specific genes/hot spots are used for treatment decision. A PCR amplicon-based enrichment strategy underlying gene panels has several benefits for clinical utility. Here, it can be mentioned its low requirement of input DNA, fast resolution time, application to FFPE and the ability to reach greater sequencing depth [62,67,110]. In addition, this strategy can handle multiple samples (patients) in one workflow. Furthermore, assumptions of primary tumor being homogenous holds little promise, as it has been shown that a primary tumor often harbors subclones of heterogenous and/or evolutionary origin [111]. Therefore, close collaboration with pathologist is important for obtaining the right tissue for NGS analysis. Reaching greater sequencing depth allows to potentially explore low frequent mutations in low tumor cellularity and/or in subclones. This identification might contribute to the refinement of diagnosis, clinical management and/or prognosis, owing to knowledge of drug-resistance before initial therapy [112]. A critical element of variant detection is the accuracy and reproducibility of the identified variants called. Hence, evaluation and validation of tools/pipelines must be compared to clear cut variants from previously well-defined samples [106].

The rate of false positives can be handled both by the specimen examined and the filters applied during bioinformatic analyses, minimizing its effects. FFPE samples harbor thousands of artifacts [113], which may remove low-frequency true variants during filtering. It is therefore crucial to consider the type of specimen, in order to deploy more accurate filters and the importance of validating the pipeline. Many studies have subjected aligners and variant callers concerning their performance, concluding limited concordance [6,57]. Filters embedded in aligners focusing on mismatches can be adjusted to allow or exclude fewer or more mismatches during alignment. With a greater number of mismatches, it also increases the likelihood of DNA fragment to align to regions with similarity. Thus, the increment of false-positive findings throughout variant calling. One the other hand, narrowing alignment filters to only accept few mismatches will potentially leave out true-positives variants with greater amounts of mismatches. Hard filters can be adjusted in most alignment and variant calling -tools to deal with difficult regions (homopolymeric and repetitive-regions). This is solved by either completely rejecting variants in these areas or apply manual empirical filters such as thresholds of coverage, Phred-score and *p*-values [114]. It is important to note that increasing or decreasing filters identifying less or additional variants in a clinical setting is not necessarily beneficial for the patient. If the identified additional variants are artifacts in relevant genes, then these could potentially lead to misguidance of therapeutic course. Moreover, an extra challenge in the clinical setting is the handling of germline mutations. Tumor samples are usually examined without a germline counterpart (i.e., normal tissue). Hence, by applying normal-like sample within the same patient, may help to reduce the number of germline variants.

Special caution must be taken setting up and interpreting NGS analysis from clinical data, as many factors can interplay with the outcome. Many answers can be intriguing, but the right answer is the one beneficial for the patient outcome. Hence, critical validation of pipelines for clinical utility is of utmost importance.

## 7. Conclusions

The advent of NGS greatly improved the study of genetics, as well as the diagnosis and treatment of genetic diseases. However, with the ability to sequence billions of reactions in a short period of time creates a demand for analytic tools to overcome these large data sets generated. Robust pipelines for NGS analysis are in constant demand, thus alignment tools and variant calling tools are still improving and are an active area of research. From the literature it has been reported that the combination of alignment tools and variant calling tools tends to vary in consistency. Interestingly, we find that from 10 studies subjecting sample NA12878, BWA-MEM or Bowtie2 in combination with Strelka2 or Mutect2 are among the best performing pipelines for SNV detection. CNV and SV detection are challenging due to the duplications/deletions of regions within a gene and/or translocation. The read-assembly approach is promising for detecting CNV/SV. Nevertheless, that requires longer spans of reads than currently provided by the present NGS technology and extensive computational resources to function ideally. The ongoing improvements to decrease the high error-rate underlying the TGS-technologies so far, will preferably solve the problem with longer reads.

The NGS technologies are increasingly being implemented into diagnostic routine settings, along with a diversity of bioinformatic. Different specimens are subjected for sequencing, according to their purpose and origin. To provide an optimal answer in patient’s course of disease, precise annotation of mutations is mandatory. Hence, the prerequisite of evaluating and validating bioinformatic pipelines used for the analysis. Laboratories conducting NGS should as a minimum participate in quality trials as documentation for applied competence conducting NGS analysis. Furthermore, consideration concerning the usage of NGS in relation to the timepoint of treatment should be taken into account.

Evidence-based biological treatment can optimally be supported by using panel sequencing (TES and/or hot-spot) ensuring fast throughput, focused datamining and high sensitivity and specificity. In contrast to WES, that can be used for exploring and prioritizing new relevant drug targets across disease used for experimental treatment of patients. However, Opposite, WGS contributes to large amounts of data, whereas 99% is information about non-coding regions. Due to the large amount of data in WGS results may be presented with low sequencing depth. Hence, the risk of not identifying relevant actionable clinical targets. Although, WGS is still beneficial of gaining new knowledge from research studies, that might benefit future patient.

## Figures and Tables

**Figure 1 cancers-13-01751-f001:**
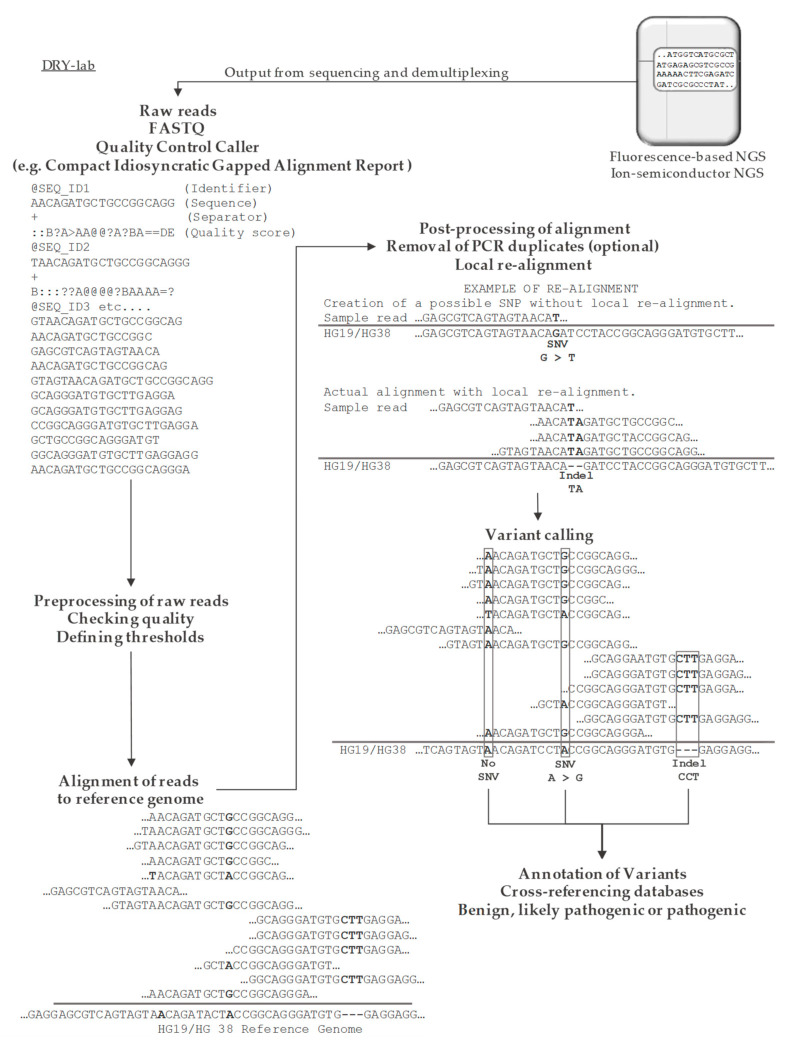
Schematic representation of the overall steps in the workflow of NGS analysis.

**Table 1 cancers-13-01751-t001:** Sequencing platforms & comparison.

Platform	Immobilization	Amplification	Sequencing Technology	Limitations & Error Rate	Read Length (bp *)	Run Time (h **)	Output (Gb ***)
1st generation technologies							
Sanger	N/A	PCR with dNTPs and ddNTPs	Irreversible chain termination	0.001%	≤900	~4	≤0.002
Maxam-Gilbert	N/A	N/A	Chemical termination of ^32^P labeled ssDNA	0.001%	≤900	N/A	≤0.002
2nd generation technologies							
HiSeq2000	Flow cell	Bridge amplification	Cyclic reversible dye chain termination	GC-rich regions 0.2%	≤125	7–144	≤1600
MiSeq	Flow cell	Bridge amplification	Cyclic reversible dye chain termination	GC-rich regions 0.2%	≤300	4–55	≤15
Ion Torrent PGM	Bead emulsion	Emulsion PCR	Synthesis depended H^+^ detection	Homo-polymers 1% Indel	≤400	2–7.5	≤2
Ion Torrent S5XL	Bead emulsion	Emulsion PCR	Synthesis depended H^+^ detection	Homo-polymers 1% Indel	≤600	2.5–4	≤25
3rd generation technologies							
ONT MinION	Processive enzyme	N/A	Monitoring the current of a nucleotide in ssDNA	5–20%	10,000–30,000	Real time	≤25
PacBio SMRT	DNA attachment to the bottom of each Zero Mode Waveguide	N/A	Detection of incorporation of fluorescent nucleotides during real time synthesis	10–15%	10,000–30,000	Real time	≤4

* bp = base pair, ** h = hours, *** Gb = gigabyte.

**Table 2 cancers-13-01751-t002:** Examples of genetic aberrations in cancers to guide personalized medicine. The list is devised from information collected from COSMIC, ClinVar and OncoKB.

Gene	Aberration	Targeting Drug	Cancer Type
*BRCA-1/2*	Loss-of-function	PARP-inhibitor	Breast cancer, Ovarian cancer, Prostate cancer
*ERBB2/HER2*	Amplification	Dimerization-inhibitor of HER2-HER3	Breast cancer
*PIK3CA*	Gain-of-function	PIK3 kinase-inhibitor	Breast cancer
*BCL2*	Gain-of-function (17 bp deletion)	Blocker of Bcl-2	Chronic lymphocytic leukemia
*RAS*	Wild type	EGFR-inhibitor	Colorectal cancer
*c-KIT*	Gain-of-function (exon 9, 11, 13, and 17)	Tyrosine kinase-inhibitors	Gastrointestinal Stromal Tumor
*EGFR*	Gain-of-function (exon 19 deletion and/or L858R)	Tyrosine kinase-inhibitors	Lung cancer, Brain cancer
*BRAF*	Gain-of-function *(V600E/K)*	Kinase-inhibitor	Melanoma
*CDK12*	Loss-of-function	PARP-inhibitor	Prostate cancer

**Table 4 cancers-13-01751-t004:** Variant calling tools.

Variant Calling Tools	Variant Detection	Latest Version	Ref.
Atlas2 suite	SNV, indels	v1.4.1	[77]
CONTRA	CNV, SV	v2.0.8	[78]
CNVnator	CNV, SV	v0.4	[79]
CoNVEX	CNV, SV	N/A	[80]
DeepVariant	SNV, indels	v1.0	[81]
DELLY	CNV, SV	v0.8.7	[82]
ExomeCNV	CNV, SV	v1.4	[83]
FreeBayes	SNV, indels	v1.3.4	[84]
GATK Haplotype Caller (GATK-HC)	SNV, indels	v4.1.9.0	[85,86]
GlfSingle	SNV, indels	N/A	N/A
ISAAC Variant Caller (IVC)	SNV, indels	V2.0.13	[74]
LUMPY	CNV, SV	v0.3.1	[87]
Magnolya	CNV, SV	v0.15	[88]
Mutect	SNV, indels	v1.1.5	[89]
Mutect2	SNV, indels	v4.1.9.0	[4]
Pindel	CNV, SV	N/A	[90]
Platypus	SNV, indels, SV	N/A	[91]
SAMtools	SNV, indels	v1.11	[92]
SNPSVM	SNV	N/A	[93]
SomaticSniper	SNV	v1.0.5.0	[94]
SpeedSeq	SNV, indels	v0.1.2	[95]
Strelka	SNV, indels	N/A	[96]
Strelka2	SNV, indels	v2.9.10	[5]
SVMerge	CNV, SV	v1.2	[97]
Torrent Variant Caller (TVC)	SNV, indels, SV	v5.12.0	N/A
Ulysses	CNV, SV	v1.0	[98]
Varscan2	SNV, Indel	v2.4.4	[99]

**Table 5 cancers-13-01751-t005:** Overview of alignment tools and variant calling tools in research papers subjecting sample NA12878.

Research Paper	Subjected Sample	Reference Genome	Alignment Tool	Variant Calling Tool
Chen et al. 2019 [61]	NA12878	WES WGS	BWA-MEM	GATK-HC SAMtools Strelka2 VarScan2
Chen et al. 2020 [62]		WES	BWA-MEM	Mutect2 Strelka2
Cornish et al. 2015 [57]		WES	Bowtie2 BWA_MEM CUSHAW3 MOSAIK Novoalign	SAMtools SNPSVM
Hwang et al. 2015 [64]		WES WGS	Bowtie2 BWA-MEM Novoalign	FreeBayes GATK-HC SAMtools TVC
Hwang et al. 2019 [63]		WES WGS	Bowtie2 BWA-MEM GSNAP ISAAC Novoalign SOAP2	Atlas2 FreeBayes GATK-HC glfSingle IVC Platypus SAMtools suite VarScan2
Kumaran et al. 2019 [6]		WES	Bowtie2 BWA-MEM MOSAIK Novoalign SOAP2	GATK-HC DeepVariant FreeBayes SAMtools
Meng et al. 2018 [65]		TES WES WGS	BWA-MEM	DeepVariant Lancet Strelka2 VarScan2
Pan et al. 2019 [59]		WGS	Bowtie2 BWA-MEM ISAAC Novoalign	FreeBayes GATK-HC IVC SAMtools
Supernat et al. 2018 [66]		WES WGS	BWA-MEM	DeepVariant GATK-HC SpeedSeq
Xu et al. 2014 [67]		WES	BWA-MEM	Mutect SomaticSniper Strelka VarScan2

## Data Availability

Not applicable.

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
