# Peer review of "Next Generation Sequencing Technology in the Clinic and Its Challenges"

_cancers, 2021, doi:10.3390/cancers13081751_

Round 1
Reviewer 1 Report
This paper provides a clinical meaningful review of the NGS analysis flow for genomic analysis. They mainly focus on the sequencing, alignment, and variant calling. This review is very informative, but there are a few comments that the authors could improve on.
1. The Table2 looks poor. It would be better to include genes at higher levels of evidence such as EGFR mutations in lung cancer. This reviewer guess you could refer to OncoKB and others.
2. The authors describe alignment and variant calling in the 5.3 Bioinformatics section. The reader may detect CNAs as well as SNV in the genetic profiling testing. In general, the detection of CNAs by NGS is more difficult than SNV detection, and the method needs improvement. A description of how to detect CNA would be very useful to the reader.
Reviewer 2 Report
The paper is an interesting review on next generation sequencing in cancer and personalized medicine.
I recommend publication subject to a minor revision based on the comments below:
- I suggest revision of English language. There are some typos all over the text.
- Some acronyms are not explained (e.g. PCR line 63).
- Lines 68-70, I would suggest to put a brief description of how NGS technology works and also on the main features of Illumina and Ion Torrent, together with the corresponding citations. Otherwise Table 1 is hard to understand for a non-expert reader.
- Table 1: please explain the terms in parenthesis (bp, h and gb).
- Figure 1 is a little confusing. It could be better ordered in order to obtain a clearer scheme.
- Figure 2 makes it difficult to follow the links with the other elements of the graph.
- The conclusion section can be extended with a summary of the previous paragraphs.
